# Infection by *Salmonella enterica* Serovar Typhimurium DT104 Modulates Immune Responses, the Metabolome, and the Function of the Enteric Microbiota in Neonatal Broiler Chickens

**DOI:** 10.3390/pathogens11111257

**Published:** 2022-10-29

**Authors:** Danisa M. Bescucci, Tony Montina, Valerie F. Boras, G. Douglas Inglis

**Affiliations:** 1Lethbridge Research and Development Centre, Agriculture and Agri-Food Canada, Lethbridge, AB T1J 4B1, Canada; 2Department of Chemistry and Biochemistry, University of Lethbridge, Lethbridge, AB T1K 3M4, Canada; 3Southern Alberta Genome Sciences Centre, University of Lethbridge, Lethbridge, AB T1K 3M4, Canada; 4Chinook Regional Hospital, Alberta Health Services, Lethbridge, AB T1J 1W5, Canada

**Keywords:** *Salmonella enterica* serovar Typhimurium, broiler chickens, immune response, enteric microbiota, metabolome, metabolomics

## Abstract

*Salmonella enterica* serovar Typhimurium incites salmonellosis in many different species including chickens and human beings. Acute salmonellosis was studied in neonatal broiler chicks by orally inoculating 2-day-old chicks with *S*. Typhimurium DT104. The temporal impact of disease (1, 2, and 4 days post-inoculation) on the structure and function of the enteric microbiota, on the bird’s immune response in the ileum, cecum, and colon, and on the metabolome of digesta, breast muscle, liver, serum, and hippocampus were examined. Substantive histopathologic changes were observed in the small and large intestine, including the colon of chicks inoculated with *S*. Typhimurium, and increased in magnitude over the experimental time period. A variety of inflammatory genes (*IFNγ*, *IL8*, *IL10*, *INOS*, *MIP1β*, *TGFβ2, TLR4*, and *TLR15*) were temporally regulated. In addition, the metabolome of ileal digesta, breast muscle, liver, serum, and hippocampus was temporally altered in infected chicks. Although the structure of bacterial communities in digesta was not affected by *S*. Typhimurium infection, metabolomic analysis indicated that the function of the microbiota was changed. Collectively, the study findings demonstrate that infection of neonatal chicks by *S*. Typhimurium imparts a temporal and systemic impact on the host, affecting the immune system, the metabolome, and the function of the enteric microbiota.

## 1. Introduction

Salmonellosis is a foodborne zoonotic disease with a variety of clinical presentations in human beings [1]. From 2006 to 2019, >96,000 cases of *Salmonella* infection were reported in Canada, and *Salmonella enterica* serovar Typhimurium was the second most reported serovar [2]. Salmonellosis is a reportable disease in Canada, and chicken products contaminated by the bacterium are a primary reservoir of *S. enterica* serovars infecting people [3]. Moreover, salmonellosis in chicks results in serious losses to the industry in terms of mortality and morbidity, including reduced bird growth and egg production [3]. In chickens, *S.* Typhimurium can incite enterocolitis that can develop into systemic infection [4]. Clinical presentation involves heterophil infiltration of the mucosa and lamina propria of the ileum and cecum, with an extended mucosa associated immune response [5] that can result in severe systemic disease [6]. Studies conducted to evaluate host-responses and pathogenic alterations have focused entirely on the ileum and cecum, disregarding impacts within the colon. The colon and rectum of chickens are often overlooked due to their short length (i.e., ≈10 cm in length in adult chickens), and the rapid transit of digesta [7]. However, co-transport of glucose and amino acids with Na^+^ is carried out in the colon [8,9]. Additionally, the expression of Toll-like receptors (TLRs) in the enterocytes of the large intestine, such as TLR1, TLR2, TLR3, TLR4, TLR6, and TLR15, demonstrates a high level of activity involving pathogen surveillance [10]. Furthermore, the main products of fermentation, such as short chain fatty acids (SCFAs), are observed in high concentrations in the colon [11].

The enteric microbiota plays an essential role in nutrient uptake, immune development, and intestinal defense via direct or indirect competition with pathogens (i.e., colonization resistance) [12]. Unlike other animals, the diversity of the enteric microbiota in newly hatched chicks is very low. The limited exposure to a maternal microbiota, and the clean conditions that hatcheries maintain, result in an intestinal bacterial community that is incapable of conferring strong protection against enteric pathogens [13]. In the first days of life, the establishment and shaping of the intestinal microbiota commences through exposure to extrinsic factors such as food, water, bedding, and the other environmental matrices. Other factors, such as gender, age, and breed have also been shown to play an important role in shaping the microbiota of chickens [14,15]. Exposing chicks at an early age to *S. enterica* serovar Enteritidis has been shown to interfere with the development of the enteric microbiota, affecting the diversity and the structure of the bacterial community, with a disproportionate abundance of bacteria within the *Enterobacteriaceae* [16]. Alterations to the enteric microbiota at such an early age, coupled with the effects of infection, adversely affects chicken performance throughout the production cycle resulting in significant production losses [17]. The impact of *S.* Typhimurium on the structure and function of the intestinal microbiota of neonatal chicks has yet to be examined.

The study of the metabolome is used to better understand host responses. Evaluation of the metabolome of chicken blood, tissues, and intestinal digesta can provide important information on the regulation of metabolic pathways, and the core metabolites associated with the different sample types [18]. Additionally, detecting changes in specific metabolites during pathogenesis can identify specific biomarkers of disease. These biomarkers can then be used as a diagnostic tool for early detection of asymptomatic infectious diseases, and to objectively evaluate mitigation strategies. A limited number of studies evaluating alterations in the metabolome of chickens following infection with *S. enterica* have been conducted to date. In this regard, alterations in cecal metabolic pathways related to arginine, proline, and tricarboxylic acid cycles were recently described in layer chickens infected with *S. enterica* serovar Enteritidis [19]. Phosphorylation changes in the 5′ adenosine monophosphate-activated protein kinase (AMPK), and pathway modification in insulin/mammalian target of rapamycin (mTOR) have also been reported in chicken muscle after 3 weeks of infection with *S.* Typhimurium [20].

To fully understand and characterize enteric disease, it is fundamental to examine the overall host response triggered by the pathogen and integrate those changes with the intestinal microbiota. In the current study, we hypothesized that early infection with *S.* Typhimurium will temporally modulate: (i) the host immune response throughout the intestine; (ii) the host metabolome; and (iii) both the structure and function of the intestinal microbiota. To test these hypotheses, 2-day-old chicks were orally inoculated with a virulent pathovar of *S.* Typhimurium (DT104), and local and systemic host responses were measured at 1, 2 and 4 days-post inoculation (dpi). Histopathologic changes and expression of mRNA were evaluated along the intestinal tract, including the colon. The systemic effects of salmonellosis were characterized using metabolomics (i.e., breast muscle, liver, serum, and hippocampus). In addition, the impacts of disease on the structure and function of the enteric microbiota were determined.

## 2. Results

### 2.1. Salmonella Typhimurium Was Observed Only in Broiler Chicks Inoculated with the Pathogen

*Salmonella* Typhimurium was not isolated from the cloaca of chicks upon arrival at the Lethbridge Research and Development Centre (LeRDC). Moreover, the pathogen was not detected in digesta or associated with the mucosa of control (CON) treatment chicks. In *S*. Typhimurium (SAL) treatment chicks, *S.* Typhimurium was not detected in the liver. The densities of the pathogen in SAL treatment chicks were higher in the digesta of cecum (*p* ≤ 0.001) and colon (*p* ≤ 0.002) relative to the ileum at 1 dpi (Appendix A) and 4 dpi (Appendix A). At 2 dpi, higher densities of *S.* Typhimurium were observed in cecal digesta (*p* ≤ 0.001), but not in colonic digesta (*p* = 0.534) relative to digesta from the ileum (Appendix A). Higher densities of the pathogen were associated with the mucosa of cecum (*p* ≤ 0.054) and colon (*p* ≤ 0.043) relative to the ileum at 1 dpi (Appendix A) and 4 dpi (Appendix A). At 2 dpi, higher densities of the pathogen associated with mucosa within the cecum (*p* = 0.003), but not mucosa in the colon (*p* = 0.492) were observed (Appendix A). No differences (*p* ≥ 0.175) were observed in densities of *S.* Typhimurium in digesta or associated with mucosa over time.

### 2.2. Infection by Salmonella Typhimurium Induced Temporal Changes in the Health Status of Broiler Chicks

Broiler chicks infected with *S.* Typhimurium did not show signs of disease at 1 or 2 dpi. Three inoculated chicks developed bloody diarrhea at 4 dpi. Although not statistically significant (*p* ≥ 0.123), a trend for lower body weight was observed in infected animals by 4 dpi (Appendix A). A trend for lower body weights was most pronounced in infected chicks exhibiting diarrhea as compared to infected chicks that did not show signs of salmonellosis (data not shown).

### 2.3. Infection by Salmonella Typhimurium Induced Temporal and Spatial Histopathologic Changes in the Intestine

Rapid and substantial histopathologic alterations were observed in SAL treatment chicks relative to CON treatment chicks. At 1 dpi, infected birds showed immune cell infiltration of the mucosal layer. Infiltration increased over time, involving the mucosa, submucosa, and muscularis layer by 4 dpi (Appendix A). Significant differences (*p* ≤ 0.001) were observed in total histopathologic scores between SAL and CON treatment animals in the ileum, cecum, and colon at 2 dpi and 4 dpi (Figure 1A–C). At 1 dpi, no histopathologic differences (*p* ≥ 0.114) were observed in the ileum and cecum between the two treatments. However, colonic tissue of SAL treatment chicks showed higher histopathologic change scores (*p* ≤ 0.001) relative to CON treatment birds at the 1 dpi endpoint.

### 2.4. Expression of Immune Genes Was Temporally Altered following Infection by Salmonella Typhimurium

Expression of hallmark genes associated with an intestinal immune response were examined at 1 dpi, 2 dpi, and 4 dpi in the ileum, cecum, and colon. SAL treatment chicks exhibited higher concentrations of interferon-γ (*IFNγ*) mRNA in the ileum (*p* = 0.025) and cecum (*p* = 0.028) at 2 dpi (Appendix A). Higher quantities of mRNA for this cytokine were also observed in SAL treatment chicks in the ileum (*p* = 0.007), cecum (*p* < 0.001), and colon (*p* = 0.002) at 4 dpi (Figure 2A–C). Expression of the chemoattractant, interleukin 8 (*IL8*) and macrophage inflammatory protein 1β (*MIP1β*) were also evaluated. SAL treatment chicks showed a higher concentrations of *IL8* mRNA in the ileum (*p* ≤ 0.015), cecum (*p* ≤ 0.040), and colon (*p* ≤ 0.001) at 2 dpi (Appendix A) and 4 dpi (Figure 2A–C). A higher concentration (*p* = 0.044) of *MIP1β* mRNA was only observed in the colons of infected chicks at 4 dpi (Figure 2C). Heterophil and macrophage activation was evaluated by measuring the expression of inducible nitric oxide synthase (*INOS*). SAL treatment chicks showed higher concentrations of *INOS* mRNA in the ileum (*p* ≤ 0.004), cecum (*p* ≤ 0.044), and colon (*p* ≤ 0.002) at 2 dpi (Appendix A) and 4 dpi (Figure 2A–C). Expression of the pattern recognition receptors, Toll like receptor (*TLR*)4 and *TLR15* were also measured. Concentrations of *TLR4* mRNA were higher in SAL treatment chicks in the ileum (*p* = 0.023) and colon (*p* = 0.028) at 4 dpi (Figure 2A,C). At 2 dpi and 4 dpi, concentrations of *TLR15* mRNA were higher in the ileum (*p* < 0.001) and colon (*p* ≤ 0.006) of SAL treatment chicks (Figure 2A,C and Appendix A). We also examined the expression of the T regulatory cytokines, *IL10* and *TGFβ2*. For these two cytokines, only concentrations of *IL10* mRNA differed (*p* = 0.018) between treatments in the ileum, and only at 4 dpi (Figure 2A). Activity of the Th17 immune response was evaluated by measuring the expression of *IL17*. Concentrations of *IL17* mRNA showed a similar pattern to other immune markers, with higher mRNA concentrations in the ileum (*p* ≤ 0.003) and colon (*p* < 0.001) of SAL treatment chicks at 2 dpi (Appendix A) and 4 dpi (Figure 2A,C). The expression of *IL18* was examined, and higher concentrations of *IL18* mRNA were observed in the ileum (*p* ≤ 0.032) and colon (*p* ≤ 0.013) of SAL treatment chickens at 2 dpi (Appendix A) and 4 dpi (Figure 2A,C). No significant differences were observed in the expression of immune markers of disease at 1 dpi between treatments (data not shown).

**Figure 1 pathogens-11-01257-f001:**
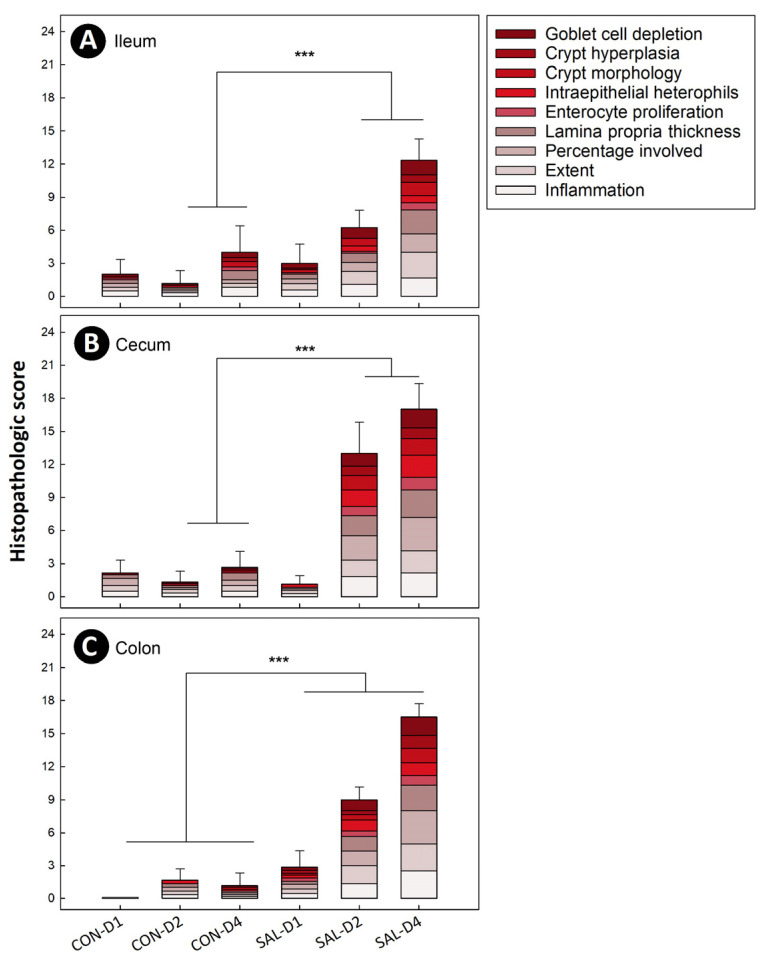
Histopathologic scores of intestinal tissues from broiler chicks orally inoculated with *Salmonella enterica* serovar Typhimurium (SAL) or medium alone (CON) at 1 (D1), 2 (D2) and 4 (D4) days post-inoculation. (**A**) Ileum. (**B**) Cecum. (**C**) Colon. Vertical lines associated with histogram bars are standard errors of the mean for the total score, and histograms bars denoted with lines and asterisks differ (*** *p* ≤ 0.001).

**Figure 2 pathogens-11-01257-f002:**
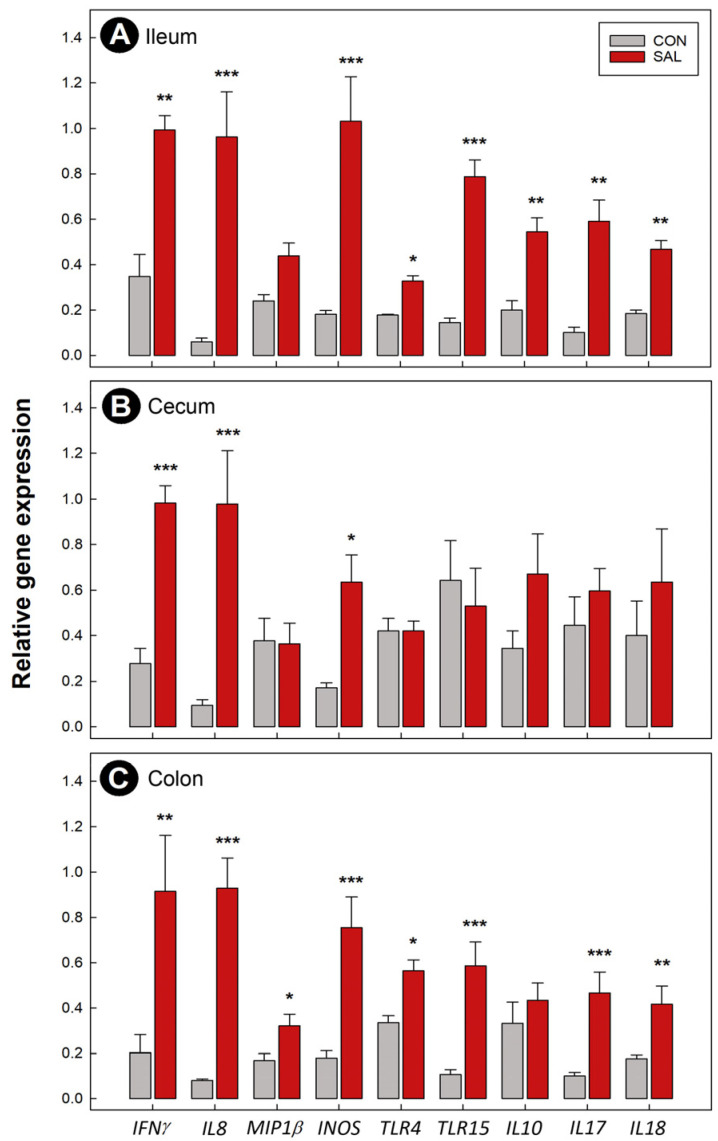
Relative expression of immune genes in broiler chicks inoculated with *Salmonella enterica* serovar Typhimurium (SAL) or administered medium alone (CON) at 4 days post-inoculation. (**A**) Ileum. (**B**) Cecum. (**C**) Colon. Vertical lines associated with histograms bars are standard errors of the mean, and histogram bars denoted with asterisks indicate differences between the SAL and CON treatments (* *p* ≤ 0.050, ** *p* ≤ 0.010, *** *p* ≤ 0.001).

### 2.5. Infection by Salmonella Typhimurium Did Not Affect Bacterial Fermentation

Concentrations of acetic, propionic, isobutyric, butyric, isovaleric, valeric, and caproic acid were evaluated in the feces of CON and SAL treatment chicks at the three experimental endpoints. Only acetic acid was found in the feces above the limit of detection. However, no significant differences (*p* ≥ 0.200) were observed in the concentration of acetic acid between the CON and SAL treatments (data not shown).

### 2.6. Infection by Salmonella Typhimurium Did Not Appreciably Affect the Structure of Enteric Bacterial Communities over Time

Characterization of bacterial communities in the ileal, cecal, and colonic digesta was carried out with next-generation sequencing (NGS). Bacterial diversity was low (Shannon’s index < 2.5), and no differences (*p* ≤ 0.109) in α-diversity were observed between CON and SAL treatment chicks in the ileum, cecum, and colon (Figure 3A–C). At all three intestinal sites, there was no difference (*p* ≥ 0.150) in β-diversity of bacterial communities between the CON and SAL treatments (i.e., as determined by Bray–Curtis and Weighted UniFrac analyses). However, the β-diversity of bacterial communities in cecal digesta changed (*p* = 0.018) between 1 dpi and 2 dpi (data not shown). A higher relative abundance of DNA from bacteria within the *Enterobacteriaceae* family was observed in cecal digesta of SAL treatment chicks at all three end points (Figure 3E). DNA of this genus was also observed at a higher relative abundance in colonic digesta (Figure 3F), but not in the ileal digesta (Figure 3D) of challenged birds at 4 dpi.

### 2.7. Infection by Salmonella Typhimurium Altered the Function of the Ileal Microbiota

Water-soluble metabolites were extracted from ileal digesta and analyzed by ^1^H-Nuclear Magnetic Resonance (NMR) spectroscopy to evaluate the impact of infection on the function of the microbiota. The metabolome was only evaluated in the digesta from the ileum at 2 dpi and 4 dpi due to insufficient digesta from other intestinal sites, and from all three sites at 1 dpi. Supervised OPLS-DA separation was observed between the ileal digesta metabolome of SAL and CON treatment chicks at 2 dpi (Q^2^ = 0.400, R^2^ = 0.868, *p* = 0.017) and at 4 dpi (Q^2^ = 0.422, R^2^ = 0.824, *p* = 0.049) (Figure 4A,B). A higher impact of *S*. Typhimurium infection was observed on the ileal digesta metabolome at 4 dpi than at 2 dpi, with a total of 36 and 8 bins differing between treatments, respectively (Appendix A). At 4 dpi, a higher (*p* ≤ 0.008) concentration of maltose, glucose, and raffinose was observed in the digesta of SAL treatment chicks, whereas the concentration of valine and isoleucine in ileal digesta of birds inoculated with *S*. Typhimurium decreased (*p* ≤ 0.004) (Figure 5A).

### 2.8. Infection by Salmonella Typhimurium Altered Metabolic Profiles of Breast Muscle, Liver, Serum, and Hippocampus

Water-soluble metabolites from breast muscle, liver, serum, and hippocampus were characterized to evaluate the systemic impacts of infection by *S*. Typhimurium on the host. The metabolome of CON treatment birds did not differ at the 1 dpi, 2 dpi, and 4 dpi endpoints; thus, CON treatment samples were grouped across sample times. At all three end points, with the exception of liver at 2 dpi, supervised OPLS-DA separation indicated that *S.* Typhimurium infection affected the metabolite profiles of breast muscle (1 dpi: Q^2^ = 0.207, R^2^ = 0.366, *p* = 0.014; 2 dpi: Q^2^ = 0.254, R^2^ = 0.604, *p* = 0.018; 4 dpi: Q^2^ = 0.604, R^2^ = 0.835, *p* = 0.015; Figure 6A–C), liver (1 dpi: Q^2^ = 0.463, R^2^ = 0.596, *p* < 0.001; 4 dpi: Q^2^ = 0.389, R^2^ = 0.706, *p* = 0.004; Figure 7A,B), serum (1 dpi: Q^2^ = 0.310, R^2^ = 0.648, *p* = 0.005; 2 dpi: Q^2^ = 0.290, R^2^ = 0.608, *p* = 0.012; 4 dpi: Q^2^ = 0.845, R^2^ = 0.914, *p* < 0.001; Figure 8A–C), and hippocampus (1 dpi: Q^2^ = 0.400, R^2^ = 0.596, *p* = 0.001; 2 dpi: Q^2^ = 0.130, R^2^ = 0.439, *p* = 0.043; 4 dpi: Q^2^ = 0.134, R^2^ = 0.326, *p* = 0.047; Figure 9A–C). For breast muscle, 8, 20 and 66 metabolite bins differed (*p* < 0.050) between the CON and SAL treatments at 1 dpi, 2 dpi, and 4 dpi, respectively (Appendix A). At 4 dpi, breast muscle of SAL treatment chicks showed higher concentrations of carnosine, adenosine monophosphate, and lactate, and lower concentrations of glucose-1-phosphate and glutamine (Figure 5B). In liver, 21, 11, and 29 bins differed (*p* < 0.050) in birds infected with *S*. Typhimurium at 1 dpi, 2 dpi, and 4 dpi, respectively (Appendix A). Higher concentrations of valine and isoleucine, and lower concentrations of glucose-1-phosphate were observed in livers of infected birds at 4 dpi (Figure 5C). Evaluation of the serum metabolome showed that 21, 12, and 19 bins differed (*p* < 0.050) between SAL and CON treatment birds at 1 dpi, 2 dpi, and 4 dpi, respectively (Appendix A). At 4 dpi, concentrations of the amino-acid glutamine decreased in the serum of SAL treatment birds (data not shown). Analysis of the hippocampus showed that 5, 13, and 6 metabolite bins differed (*p* < 0.050) between the two treatments at 1 dpi, 2 dpi, and 4 dpi, respectively (Appendix A). The most prominent impact of infection by *S*. Typhimurium on the metabolome of the hippocampus was observed at 1 dpi (Figure 9A), followed by 2 dpi (Figure 9B), with attenuation occurring at 4 dpi (Figure 9C). The only metabolite that was consistently affected in the hippocampus was glutamate, with a higher concentration of this metabolite observed in infected chicks at all three end points (data not shown).

## 3. Discussion

A number of studies have examined the impacts of infection by *S.* Typhimurium in chickens; however, the vast majority of these studies have focused on one particular aspect of the disease, either the immune response, changes in the microbiota, or the impact in the metabolome [4,5,21]. A comprehensive investigation to ascertain how *S.* Typhimurium impacts broiler chickens, particularly early in life, has not been completed. In the current study we temporally (1 dpi, 2 dpi, and 4 dpi) and spatially examined the impacts of *S.* Typhimurium infection on intestinal histopathologic changes, host immune responses, the structure and function of the microbiota, and the metabolome of various tissues, including, breast muscle, liver, serum and the hippocampus. A salient goal of the study was to advance knowledge on the *S*. Typhimurium-host-microbiota interaction in broiler chicks.

Salmonellosis is of concern to the poultry sector, not only due to the risk of human disease, but also due to the economic losses caused by mortality, and the reduction in bird performance triggered by the disease. We observed that a subset of infected chicks presented signs of diarrhea, and they exhibited lower body weights as compared to other inoculated and control treatment chicks. Even though the differences in body weight of all of the chicks inoculated with *S.* Typhimurium did not differ significantly from that of control treatment chicks, chicks inoculated with the pathogen showed a trend for lower body weights. Optimal growth during the starter period is essential, and any reduction in body weight due to infection impacts overall production performance. In this regard, Adhikari et al. [22] reported that chicks infected with *S.* Typhimurium did not show differences in body weight during the starter phase, but did show significant differences during the finisher phase. Our research group previously observed the same trend in piglets infected with *S.* Typhimurium DT104 [23] with body weight loss increasing incrementally over time.

**Figure 6 pathogens-11-01257-f006:**
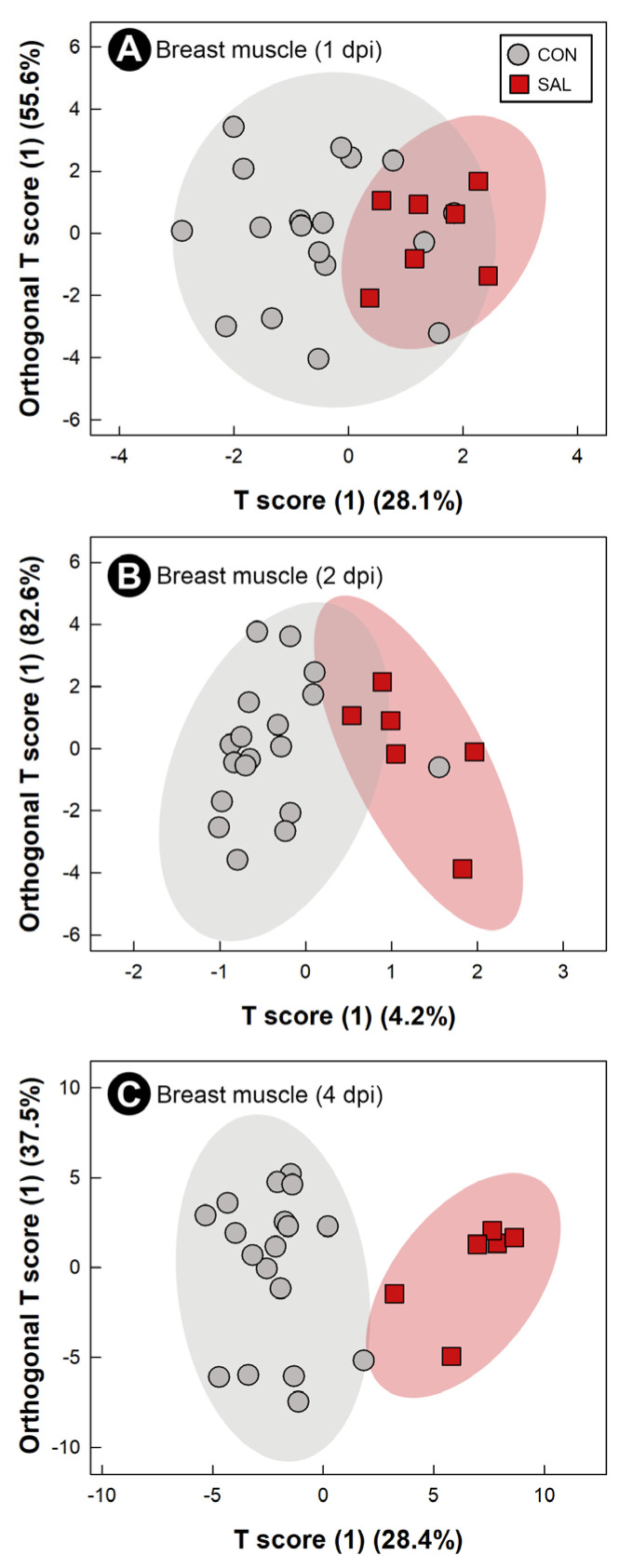
Orthogonal Projections to Latent Structures Discriminant Analysis (OPLS-DA) scores plot of breast muscle metabolites in broiler chicks inoculated with *Salmonella enterica* serovar Typhimurium (SAL) or administered medium alone (CON). (**A**) 1 day post-inoculation (dpi). (**B**) 2 dpi. (**C**) 4 dpi. Each circle or square represents one bird, and data were plotted using significant metabolites identified by Mann–Whitney U test and Variable Importance Analysis based on Variable Combination (VIAVC) machine learning. Shaded ellipses represent 95% confidence intervals.

To temporally evaluate the impact of infection by *S*. Typhimurium on broiler chicks, samples were obtained at 1 dpi, 2 dpi, and 4 dpi. Our results correspond with previously published findings [4] that reported rapid and substantial histopathologic changes. However, we observed that the severity of disease increased incrementally over time with immune cell infiltration extending from the mucosa to the muscular layer by 4 dpi. Very limited research has been conducted to examine the impact of enteric infections on the colon of chickens, despite the importance of the colon for overall bird health [11,24]. We observed that histopathologic changes within the colon of chicks inoculated with *S*. Typhimurium occurred to the same degree as was observed in the ileum and cecum. Inflammation was characterized by immune cell infiltration, crypt morphology alteration, goblet cell depletion, and an increase in intraepithelial heterophils. In contrast to the ileum and cecum, conspicuous histopathologic changes were observed in the colon as early as 1 dpi.

**Figure 7 pathogens-11-01257-f007:**
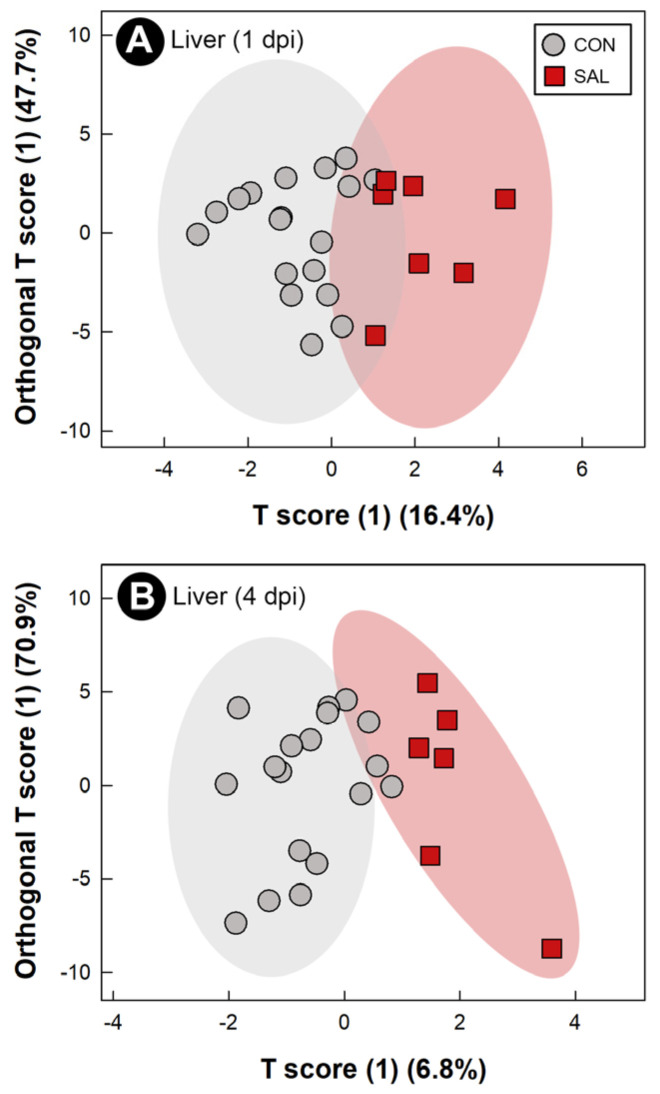
Orthogonal Projections to Latent Structures Discriminant Analysis (OPLS-DA) scores plot of liver metabolites in broiler chicks inoculated with *Salmonella enterica* serovar Typhimurium (SAL) or administered medium alone (CON). (**A**) 1 day post-inoculation (dpi). (**B**) 4 dpi. Each circle or square represents one bird, and data were plotted using significant metabolites identified by Mann–Whitney U test and Variable Importance Analysis based on Variable Combination (VIAVC) machine learning. Shaded ellipses represent 95% confidence intervals.

To further ascertain the impact of infection by *S*. Typhimurium on broiler chicks, we temporally quantified expression of immune markers of disease. A previous study showed that an immune response in the cecum of chickens inoculated with *S.* Typhimurium was triggered at 3, 5, and 7 dpi [5]. However, they did not evaluate the degree of inflammation in other intestinal sites, including the ileum and colon. We observed that an immune response was initiated along the intestinal tract (i.e., ileum, cecum, and colon), and exhibited a temporal progression. Although histopathologic changes were observed at 1 dpi in the colon, expression of immune markers were observed primarily at 2 and 4 dpi. Additionally, our study demonstrated a rapid and significant expression of cytokine and chemokine genes in both the ileum and colon. It is noteworthy that the expression of TLRs in the chicken intestine plays a pivotal role in the recognition of enteric pathogens. TLR4 is an important mediator of innate immune response activation following infection with *S.* Typhimurium via recognition of lipopolysaccharide (LPS) [25], and we observed that *TLR4* was highly upregulated in the ileal and colonic tissue of infected animals at 4 dpi.

After activation of *TLR4*, a pro-inflammatory response consisting of *IL8*, *IL17*, and *MIP1β* is triggered resulting in the recruitment of inflammatory cells to the site of infection [4,5]. Arrival of inflammatory cells, specifically macrophages and heterophils, results in an increase in reactive nitrogen species (e.g., *INOS*) [26]. Heterophils are the avian equivalent to neutrophils, playing an essential role in impairing salmonellosis incited by *S.* Enteritidis in birds [27]. The early expression of *IL8* and *IL17* that we observed corresponds with high densities of the pathogen present in the intestine. Thus, the upregulation of the cytokines in the lamina propria and muscularis layer of infected chicks is consistent with the high influx of heterophils observed. Expression of *MIP1β* was upregulated in the colonic tissue at 4 dpi, which corresponds with findings from Berndt et al. [4]. They reported that the expression of this chemotactic cytokine fluctuated over time, with the highest levels of *MIP1β* occurring at 4 dpi. Macrophage produced *INOS* can suppress *S.* Typhimurium proliferation making it an essential mechanism of protection [28]. Thus, the elevated expression of *INOS* at 2 and 4 dpi corresponds directly with the acute inflammation that we observed. Intestinal colonization by *S.* Typhimurium induces a Th1 response, with higher expression of *IFNγ* directed to eliminate the intracellular pathogen [29]. Additionally, stimulation of *INOS* production was associated with *IFNγ* release [30]. The higher expression of *IFNγ* that we observed at 2 and 4 dpi in chicks infected with *S.* Typhimurium could have been responsible for the production of INOS from macrophages. A study conducted in mice showed the importance of caspase-1 in the host immune response against *S.* Typhimurium via activation of the proinflammatory cytokine, *IL18* [31]. Higher expression of *IL18* following *S.* Typhimurium infection has also been previously demonstrated in layer chicks [4].

**Figure 8 pathogens-11-01257-f008:**
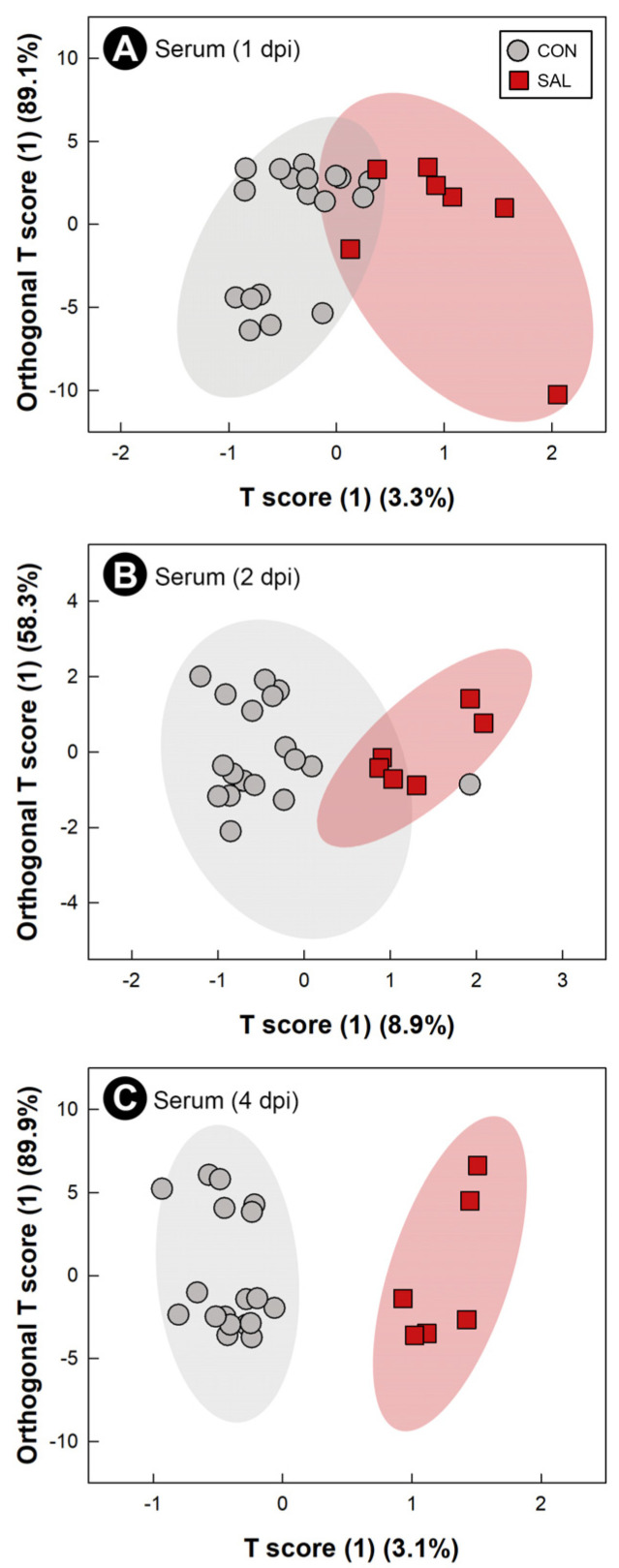
Orthogonal Projections to Latent Structures Discriminant Analysis (OPLS-DA) scores plot of serum metabolites in broiler chicks inoculated with *Salmonella enterica* serovar Typhimurium (SAL) or administered medium alone (CON). (**A**) 1 day post-inoculation (dpi). (**B**) 2 dpi. (**C**) 4 dpi. Each circle or square represents one bird, and data were plotted using significant metabolites identified by Mann–Whitney U test and Variable Importance Analysis based on Variable Combination (VIAVC) machine learning. Shaded ellipses represent 95% confidence intervals.

**Figure 9 pathogens-11-01257-f009:**
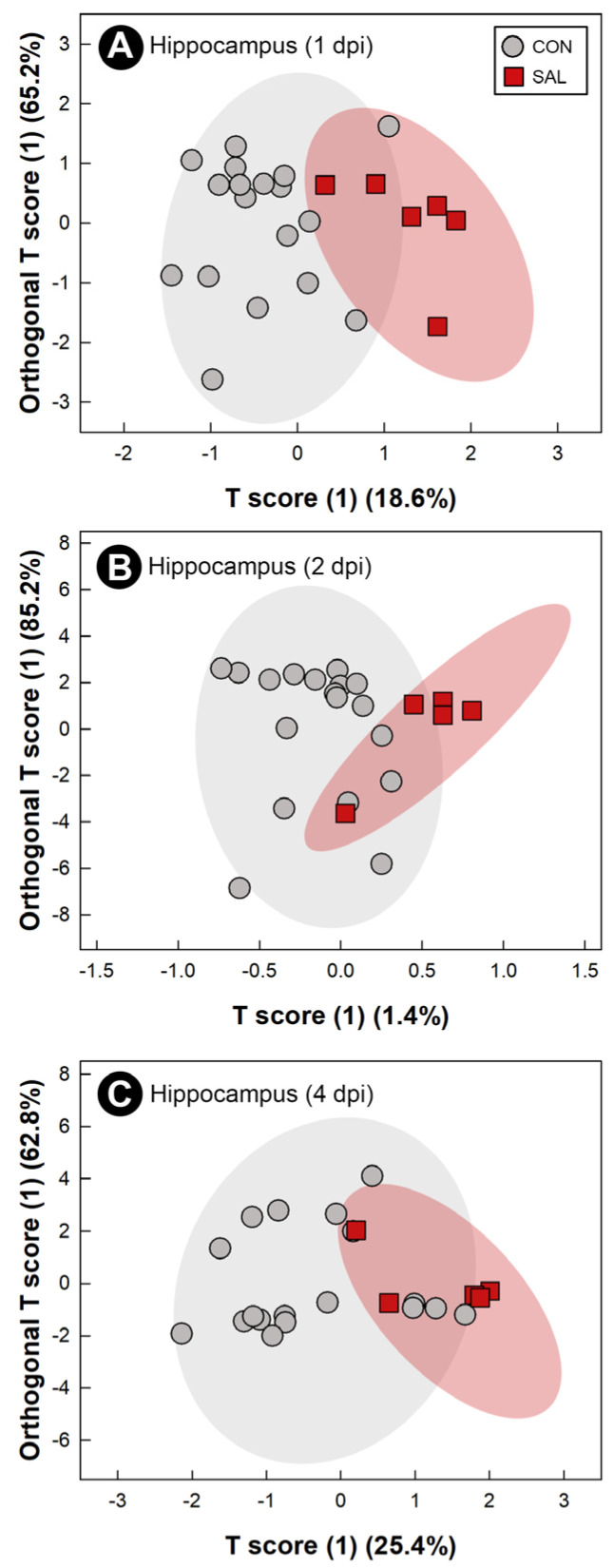
Orthogonal Projections to Latent Structures Discriminant Analysis (OPLS-DA) scores plot of hippocampus metabolites in chicks inoculated with *Salmonella enterica* serovar Typhimurium (SAL) or administered medium alone (CON). (**A**) 1 day post-inoculation (dpi). (**B**) 2 dpi. (**C**) 4 dpi. Each circle or square represents one bird, and data were plotted using significant metabolites identified by Mann–Whitney U test and Variable Importance Analysis based on Variable Combination (VIAVC) machine learning. Shaded ellipses represent 95% confidence intervals.

TLR15 has been described in chickens, with no orthologues found in human beings or mice [32]. This member of the TLR-family has been associated with infections by various pathogens including *Eimeria tenella* [33], *Mycoplasma synoviae* [34], and *S.* Typhimurium [32]. It has been previously proposed that TLR15 is functionally equivalent to TLR2, being able to recognize pathogen-associated molecular patterns such as lipoprotein and LPS [32]. Therefore, the higher expression of *TLR15* that we observed in the ileum and colon of broiler chicks at 2 and 4 dpi could be associated with the recognition of the pathogen, thereby resulting in a more robust immune response. Importantly, the triggering of an exacerbated immune response has to be closely regulated by the host to avoid extensive damage, and we observed upregulation of the anti-inflammatory cytokine, *IL10*, which corresponds with previous findings [5].

Studies conducted in other species such as mice and piglets have shown that *S.* Typhimurium targets and effectively colonizes both the cecum and the colon [35,36]. A major objective of the current study was to determine the impact that *S.* Typhimurium infection has on the colon, an area of the intestine that has not received much attention in chickens. We determined that *S.* Typhimurium infection impacts the colon of chickens to the same degree as other intestinal sites. From 1 dpi to 4 dpi, the colon presented high densities of *S.* Typhimurium, extensive histopathologic damage, and high expression of pro-inflammatory cytokines. The exacerbated immune response triggered in the colon corresponds to an active response against the pathogen, demonstrating an important involvement of the colon in the defense against the disease. It is noteworthy that the colon plays an essential role in intestinal health via metabolism of SCFAs [37]. Different studies have focused on elucidating the biological functions of SCFAs, and the production of butyrate has been shown to inhibit chicken colonization by *S. enterica* and *Clostridium perfringens* [38,39]. Additionally, butyrate has been shown to play an important role in cell growth, immune function, and intestinal health [11]. When butyrate is released, 70–80% of its metabolism occurs in the colon of people and mice [40,41]. Thus, the involvement of this intestinal site in the host response is not limited only to an immune response. In the current study, broiler chicks were purposely maintained in individually ventilated cages devoid of microorganisms to reduce any potential confounding effects of the uncontrolled introduction of bacteria, and a low diversity enteric bacterial community was maintained in chicks throughout the study period. Thus, quantification of many SCFAs, the final products of bacterial fermentation, were not observed above the limit of detection, consistent with the low diversity microbiota that we observed in the chicks [11,24]. Future experimentation should be conducted to evaluate the effect that *S.* Typhimurium infection has in the temporal metabolism of butyrate in the colon, when a more diverse microbiota is present.

Another important component of enteropathogenic diseases is the pathogen’s impact on the autochthonous enteric microbiota. Newly hatched chicks possess a low diversity microbiota that confers limited protection against pathogens [42]. During the first weeks of life, the enteric microbiota of chickens is established via contact with the environment, food, water, and bedding [14]. In the current study, characterization of the intestinal microbiota showed similar bacterial communities within the ileum and the colon. This could be the result of the digesta transit observed in chickens. It has been previously described that the majority of the intestinal contents pass from the small intestine to the colon and are excreted in feces [43]. Only a small amount of the ileal contents will pass to the cecum for fermentation, which is voided into the colon twice per day [44]. Depending on the moment of sampling, the colonic microbiota will vary, thereby representing the composition of the ileal or cecal microbiota. In the current study, the colonic microbiota mimicked that of the ileum, which could also have contributed to the low quantities of SCFAs that were observed in the feces voided by chicks.

Early exposure of chicks to pathogens can contribute to a reduction in community diversity [16], and it has been shown that inflammation incited by *S. enterica* confers a competitive advantage to the pathogen [45]. In this regard, an inflammatory response is normally characterized by the oxidative burst caused by heterophils and macrophages, which in turn is accompanied by high levels of oxygen in the lumen of the intestine [46]. Elevated levels of oxygen can benefit the growth of certain bacterial taxa (e.g., facultative anaerobes) such as *S. enterica*, over obligate anaerobic bacteria. Members of the phylum *Pseudomonadota* (synonym, *Proteobacteria*) have been observed to increase due to inflammation, which has been considered a hallmark of microbiota changes associated with dysbiosis [47]. Although neither alpha nor beta diversity differed between infected and control chicks, we observed a significant increase in bacteria within the *Enterobacteriaceae* family in the ceca and colon of chicks inoculated with *S*. Typhimurium at both 2 dpi and 4 dpi. This corresponds directly with the histopathologic changes observed at these intestinal sites. In barn settings (i.e., without the introduction of a pathogen), the diversity of the microbiota gradually increases with the age of the chick [42]. However, early colonization of an intestine possessing a low microbial diversity is advantageous to *S. enterica* [16], and the observed increase in DNA of *Enterobacteriaceae* determined by NGS was likely due to an increased abundance of *S*. Typhimurium in chicks inoculated with the pathogen.

As a result of the current limitations of technologies to characterize bacterial communities (e.g., low taxa resolution), subtle differences can be missed [48]. Moreover, the application of NGS techniques does not provide direct evidence of microbial function. To examine the potential impact of *S*. Typhimurium on the function of the enteric microbiota, metabolomic analysis of ileal digesta were conducted. Despite no detectable differences in community structure as determined by NGS, the metabolome differed significantly between control and infected chicks. This indicates that infection altered the function of microorganisms within the ileum. The elevated presence of sugar moieties (e.g., lactose, raffinose, melibiose) in the intestine of *S.* Typhimurium infected mice has been previously described [49]. Deatherage Kaiser et al. [49] suggested that this could be a result of *Salmonella* not being able to use certain monosaccharides, and the absence of other members of the microbiota capable of metabolizing them due to the inflammation. Additionally, absorption of glucose is impaired after infection of chickens with *Campylobacter jejuni* [50]. In the current study, higher concentrations of raffinose, maltose, and glucose were observed in the ileal digesta of infected animals. As indicated, the higher concentrations of these monosaccharides could be the result of either impairment of their absorption, or the absence or low abundance of commensal microorganisms able to metabolize them, or both. Enteropathogenic infections caused by *C. jejuni* [50] or *C. perfringens* [13] have been previously associated with lower concentrations of branch chain amino acids (BCAAs) (e.g., valine, leucine, isoleucine). Our research group previously demonstrated that the concentration of these amino acids in the digesta of *C. perfringens* infected chickens increased when transplanted with a complex microbiota [13]. Furthermore, BCAAs have been shown to modulate the chicken immune response [51]. Thus, the lower concentration of valine and isoleucine observed in ileal digesta of infected animals could be the result of a deficient microbiota unable to increase the concentration of these metabolites, potentially hindering the immune response against the pathogen.

The application of metabolomics has been used to characterize early host changes caused by disease [13]. In the current study, the metabolomes of breast muscle, liver, serum, and hippocampus were temporally examined to elucidate systemic changes that *S.* Typhimurium incites in the early stages of the disease. Inoculation with *S*. Typhimurium triggered a change in the metabolome, associated with the development of the pro-inflammatory response. Acute infection by *S.* Typhimurium can induce glycolysis in the host in an attempt to energetically supply the triggered immune response [20]. Higher activation of glycolysis and increased lactate production by neutrophils have also been associated with infection by *S.* Typhimurium [52]. We observed a pro-inflammatory response with a high demand for energy in every tissue that was examined. Thus, glucose-1-phosphate, which is the first metabolite utilized to initiate glycolysis [53], expectantly decreased in the liver and breast muscle of infected animals. This could indicate an active consumption of glucose-1-phosphate. Furthermore, lactate, the final product of anaerobic glycolysis, was observed at higher concentrations in breast muscle of infected animals. Another metabolite associated with an active immune response is glutamine. Glutamine is the most prevalent amino acid in the blood stream, and even though it is considered a non-essential amino acid, it has been shown to be essential during an infection [54]. In this regard, glutamine participates in lymphocyte proliferation and cytokine production, macrophage phagocytic and secretory activities, and neutrophil bacterial killing [54]. Notably, administration of this metabolite to chickens under stress improved chicken performance and meat quality [55]. Additionally, depletion of glutamine has been associated with stress conditions [56]. We observed a drop in the concentration of glutamine in breast muscle and serum at 4 dpi, and it is likely that the observed decrease in glutamine resulted from the acute inflammatory response, as well as from cellular stress. Importantly, a reduction in the concentration of glutamine has been linked to compromised meat quality [55].

An exacerbated immune response represents a high risk to the host, due to the secondary damage that can be incited. Mechanisms for suppressing or regulating inflammation are normally activated alongside the pro-inflammatory response. BCAAs have been described as modulators of the chicken immune response [51], and an increase in these amino acids has been found in the liver of mice infected with *S.* Typhimurium [57]. Our findings are similar to those observed in mice, in which high concentrations of valine and isoleucine were present in the livers of infected broiler chicks. This alteration in the liver metabolome could be indicative of the modulation of the immune response by the host. Kogut et al. [20] showed that alterations in the adenosine monophosphate (AMP) to adenosine triphosphate (ATP) ratio activates AMP-activated protein kinase, which inhibits inflammatory responses mediated by *S.* Enteritidis. Another metabolite that has been demonstrated to participate in host protection is carnosine. This dipeptide conveys protection by sequestering reactive oxygen species and reactive nitrogen species [58]. In the chicken model of acute salmonellosis reported herein, both AMP and carnosine were found at higher concentrations in the breast muscle of infected chickens, thereby suggesting that the chicks attempted to counteract the exacerbated immune response incited by *S*. Typhimurium.

The presence of metabolites associated with regulation of inflammation were also found in the hippocampus. Glutamate has been described as the primary excitatory neurotransmitter in the central nervous system [59]. It has been shown that this metabolite plays a major role in modulation of the hypothalamus-pituitary-adrenocortical (HPA) stress response [60]. Previous studies conducted in rats demonstrated that under acute stress, levels of glutamate increased in the hippocampus [61]. In the current study, we observed that the concentration of this metabolite changed in the hippocampus over time. Since high concentrations of the excitatory amino acid glutamate have been associated with stress response [62], it is not surprising that a higher concentration of glutamate was encountered in the hippocampus at 1 dpi. The acute inflammatory response could be subjecting the host to an acute stress that triggers higher concentrations of the amino acid in the hippocampus. Additionally, damage to the brain has been associated with a high concentration of glutamate [63]. Thus, the high concentration of glutamate observed at 1 dpi, followed by a lower concentration of the metabolite at 2 dpi and 4 dpi could indicate a response by the birds to avoid possible damage caused by the excitatory amino acid. 

## 4. Materials and Methods

### 4.1. Experimental Design

The study was designed as a factorial experiment with two levels of *S.* Typhimurium challenge (CON and SAL) and three sample times (1 dpi, 2 dpi, and 4 dpi) arranged as a complete randomized design with six animals per treatment (replicates) (Appendix A). The experiment was repeated on three separate occasions (i.e., “runs”) with two replicates included in each run (total 36 animals).

### 4.2. Animals and Husbandry

On the morning of hatch, Ross 308FF broiler chicks were obtained from a local hatchery (Lethbridge, AB, Canada). Chicks were placed in individually ventilated cages (Techniplast, Montreal, QC, Canada) with sterile wood bedding and allowed to acclimatize for 2 days; the cages were operated in containment mode. A custom starter diet was provided upon arrival, and chicks had ad libitum access to food and water. Temperature, lighting and humidity were maintained following the Canadian Council on Animal Care Guidelines [64]. To ascertain weight gain, birds were weighed every other day, and at endpoints.

### 4.3. Isolation of Salmonella enterica

Isolation of *S. enterica* was conducted as previously described [23]. Briefly, on the day of arrival, the cloaca of each chick was sampled with a sterile moistened swab, and the swab was placed in a 15 mL falcon tube (VWR, Mississauga, ON, Canada) containing 5 mL of buffered peptone water (Oxoid Inc., Nepean, ON, Canada). Tubes were vortexed, and the liquid was transferred to a sterile culture tube, and incubated overnight at 37 °C. An aliquot of 50 µL of the culture was transferred to 5 mL of Rappaport-Vassiliadis enrichment broth (Oxoid Inc.) and incubated at 42 °C for 16 to 24 h. A subsample of the Rappaport-Vassiliadis enrichment broth was transferred with a 10 µL inoculation loop to Brilliant Green agar (BGA; BD Difco, Mississauga, ON, Canada) and modified lysine iron agar (MLIA; Oxoid Inc.). The BGA and MLIA cultures were incubated for 48 h at 37 °C to allow H_2_S production. If red colonies on BGA were observed, colonies were transferred to triple sugar iron agar (TSIA; BD Difco) slants. If black colonies were present in MLIA, the colonies were transferred to TSIA slants. The TSIA slant cultures were incubated at 37 °C for 16 to 24 h. In cases where representative colonies were present in the TSIA slants, the colonies were transferred to MacConkey agar (BD Difco) and incubated at 37 °C for 16 to 24 h. Colorless colonies on MacConkey agar were considered *S. enterica*.

### 4.4. Inoculation of Broiler Chicks with Salmonella enterica Serovar Typhimurium

A starter culture of *S.* Typhimurium DT104 (strain SA970934) was aerobically grown overnight at 37 °C on MacConkey agar (BD Difco). Biomass was removed from the surface of the agar and transferred to 50 mL of Columbia broth (CB) (BD Difco). Cultures were shaken at 150 rpm for 180 to 210 min at 37 °C, until an optical density at 600 nm of ≥1.2 was obtained. Cultures were centrifuged for 15 min at 4000× *g*. Supernatants were removed to a volume of 15 mL, and the density of cells was adjusted to a target of 3 × 10^9^ cells/mL. To confirm the densities of viable cells, the inoculum was diluted in a 10-fold dilution series; from each dilution, aliquots of 100 µL were spread in duplicate onto MacConkey agar. After 24 h incubation at 37 °C, *S.* Typhimurium colonies were counted at the dilution yielding 30 to 300 colony forming units. Two-day-old chicks were orally gavaged with 200 µL of *S.* Typhimurium cells suspended in CB (SAL treatment) or with CB alone (CON treatment).

### 4.5. Animal Euthanasia and Sample Collection

At the three defined endpoints (1 dpi, 2 dpi, and 4 dpi), birds were anesthetized with isoflurane (5% isoflurane; 1 L O_2_/min) and humanely euthanized by cervical dislocation while under general anesthesia. Immediately after euthanasia, the thoracic and abdominal cavities were opened. Blood was collected directly from the heart, and serum was obtained for metabolomics analysis. The distal region of the small intestine (i.e., ileum), and the cecum and colon were aseptically removed and examined for gross evidence of inflammation. From these intestinal sites, digesta was collected, and snap frozen in liquid nitrogen for community analysis, metabolomics, and *S. enterica* quantification. Intestinal tissue samples were also placed in RNAlater™ Stabilization Solution (Thermo Fisher Scientific Inc., Mississauga, ON, Canada) for quantification of mRNA. Remaining intestinal tissue was sampled using the Swiss rolling technique [65], placed in TrueFlow Macrosette cassettes (Tissue Path; Thermo Fisher Scientific Inc.), and stored in 10% neutral buffered formalin (Leica Canada Inc., Concord, ON, Canada) for histopathologic analysis. Samples from breast muscle (pectoralis major) and liver were aseptically removed and immediately frozen in liquid nitrogen for metabolomic analysis. The brain was removed, and the hippocampus was aseptically recovered and immediately frozen in liquid nitrogen for metabolomic analysis; the sample included the hippocampus proper as well as part of the adjacent area parahippocampalis, which together form the avian hippocampal formation. With the exception of samples for histopathologic analysis, all samples were stored at −80 °C until processed. 

### 4.6. Quantification of Short Chain Fatty Acids

Concentrations of SCFAs in feces were determined as previously described [66]. Briefly, fresh feces (i.e., 600–800 g) were collected at endpoint days and kept at room temperature for 30 min. Feces were homogenized in phosphate-buffered saline (pH 7.2) at 1:1 (*w*/*v*). Twenty-five percent meta-phosphoric acid (Sigma Aldrich, Oakville, ON, Canada) was added to the homogenate at 1:4 (*v*/*v*), vortexed for 30 s, and centrifuged for 75 min at 16,000× *g*. Supernatants were collected and stored at −20 °C. Concentrations of acetic, propionic, isobutyric, butyric, isovaleric, valeric, and caproic acid were determined with a gas chromatograph (Model 6890N with 7683 Series Injector; Agilent Technologies Canada Inc., Mississauga, ON, Canada) as previously described [67].

### 4.7. Scoring of Histopathologic Changes

Samples for histopathologic evaluation were processed as previously described [55]. Briefly, samples were dehydrated, embedded in paraffin, sectioned at 5 µm, deparaffinized with xylene, and stained with hematoxylin and eosin. Histologic sections were scored by board certified pathologist (V.F.B.) who was blinded to treatments, using modified previously described scoring systems [66,67,68,69]. Scoring parameters included inflammation (0–4), inflammation extent (0–4), percent involved with inflammation (0–4), lamina propria thickness (0–3), enterocyte proliferation (0–3), intraepithelial heterophils (0–3), villar height (0–3), crypt morphology (0–3), villus height to crypt depth ratio (0–3), crypt hyperplasia (0–3), and goblet cell densities (0–3). The total histopathologic score was calculated by summing the scores of all categories (maximum score of 36). Histopathologic data were analyzed using the pairwise Fisher’s exact test in R [70].

### 4.8. Quantification of Immune Gene mRNA

RNA extraction from ileum, cecum and colonic tissues was conducted with RNeasy Plus Mini Kit (Qiagen Inc., Toronto, ON, Canada) following the manufacturer’s protocol. RNA quality and quantity was assessed using a Bioanalyzer 2100 (Agilent Technologies Canada Inc.). RNA (1000 ng) was converted to cDNA with a QuantiTect Reverse transcription kit (Qiagen Inc.). Quantitative PCR reactions were run using a QuantStudio 6 (Thermo Fisher Scientific Inc., Waltham, MA, USA), and quantities of *IFNγ*, *IL8*, *IL10*, *INOS, MIP1β*, *TGFβ2, TLR4*, and *TLR15* mRNA were measured. Each reaction contained 5 µL of PerfeCTa SYBR (QuantaBio, Beverly, MA, USA), 0.5 µL of each primer (10 µM), 3 µL of RNase-free water, and 1 µL of cDNA. Cycle conditions were: 95 °C for 15 min; 40 cycles of 95 °C for 15 s, 55–60 °C for 30 s, and 72 °C for 30 s. Primer sequences specific to gene targets were obtained from the literature (Appendix A). A melt curve analysis was conducted to ascertain the specificity of amplification (55 to 95 °C). Each reaction was run in triplicate, and the mean of the three observations was used. Concentrations of cDNA were normalized relative to reference genes, β-actin and glyceraldehyde 3-phosphate dehydrogenase (GAPDH) using qBase+ software (Biogazelle, Gent, Belgium) [68]. Gene expression data was log_10_ transformed to achieve normality. Two-way ANOVA to evaluate differences among experimental factors was conducted in R [69]. In the event of a main effect (*p* ≤ 0.050), the least squares means test was used to compare treatments.

### 4.9. Quantification of Salmonella enterica Serovar Typhimurium

DNA from ileal, cecal, and colonic digesta was extracted using QIAmp PowerFecal Pro DNA kit (Qiagen Inc.) according to manufacturer’s protocol. DNA from ileal, cecal, colonic, and liver tissues was extracted using the DNeasy Blood and Tissue kit (Qiagen Inc.). To quantify *S.* Typhimurium by qPCR, duplicate reactions were prepared, with each containing 10 µL of QuantiTect SYBR Green Master mix (Qiagen Inc.), 1 µL of forward and reverse primers (0.5 µM) (IDT, San Diego, CA, USA), 2 µL of DNA, 2 µL of BSA (Promega, Madison, WI, USA), and 4 µL of nuclease free water (Qiagen Inc.). Primers used were F-(Sal) and R-(Sal) [70]. Data was collected using Mx3005p Realtime PCR instrument (Agilent Technologies Canada Inc.) with the following cycle conditions: 95 °C for 15 min, 40 cycles of 94 °C for 15 s, 64 °C for 30 s, and 72 °C for 30 s. A melt curve analysis was conducted (55–95 °C). A standard curve was prepared with serial dilutions of genomic DNA (6.72 × 10^6^ copies/µL) extracted from pure cultures of the *S*. Typhimurium. *Salmonella enterica* copy number data was log_10_ transformed to achieve normalization. Two-way ANOVA was conducted in R [69] as described above.

### 4.10. Characterization of Bacterial Communities

DNA extracted from ileal, cecal, and colonic digesta was subjected to a dual-indexing sequencing strategy to create a 16S rRNA libraries of the V4 region. Extracted DNA was first amplified with Illumina indexed adaptor primers (V4 Schloss primers) [71]. PCR reactions were conducted on an Eppendorf Mastercycler Pro S thermocycler (Eppendorf Canada Ltd., Mississauga, ON, Canada). The reactions contained 12.5 µL of Paq5000 Master Mix (Agilent Technologies Canada Inc.), 1 µL of forward and reverse primers (10 µM) (IDT, Coralville, IA, USA), 8.5 µL of Nuclease Free Water (Qiagen Inc.), and 2 µL of DNA. The following cycle conditions were applied: 95 °C for 2 min, followed by 25 cycles of 95 °C for 20 s, 55 °C for 15 s, and 72 °C for 5 min, and one cycle at 72 °C for 10 min. Purification of amplicons was achieved with 24 µL of AMPure XP beads (Beckman Coulter Canada Inc., Mississauga, ON, Canada) following the protocol provided by manufacturer. Purity and size of the amplicons were confirmed by Agilent High Sensitivity DNA chips used on a Bioanalyzer 2100 (Agilent Technologies Canada Inc.). Amplicon quantification was conducted using a Qubit 4 fluorometer (Thermo Fisher Scientific Inc.). DNA libraries from each sample were normalized to 1.5 ng/µL, and pooled together for a final library of 4 pM. PhiX control DNA (Illumina Inc., San Diego, CA, USA) was added at 25% as a sequencing control. Both PhiX and the library were denatured and diluted to a final concentration of 4 pM prior to loading into a MiSeq Reagent Kit v2 500-cycle. Sequencing was conducted using an Illumina MiSeq platform (Illumina Inc.). Analysis of data was conducted in QIIME2 [72] version 2021.11. DADA2 [73] was used for filtering low quality reads, trimming sequences, and for paired joining of forward and reverse reads. Clustering of reads into exact Amplicon Sequence Variants (ASVs) was conducted with a q2-feature classifier using the SILVA bacteria reference database. Alpha and beta-diversity analyses were also conducted in QIIME2. Alpha-diversity differences were analyzed by pairwise comparisons of Kruskal–Wallis test. Beta-diversity was subjected to pairwise permutational multivariate analysis of variance (PERMANOVA). Relative abundances of ASVs were converted to log ratios and differential abundance analyses were evaluated with ANCOM in QIIME2.

### 4.11. Characterization of the Metabolome

Metabolomics data was collected using a 700 MHz NMR spectrometer (Bruker Avance III HD NMR spectrometer; Bruker, Milton, ON, Canada) as previously described [24]. In brief, ileal digesta (150 mg), breast muscle (150 mg), liver (150 mg), serum (150 µL), and hippocampus (150 mg) were suspended in metabolomics buffer (0.125 M KH_2_PO_4_, 0.5 M K_2_HPO_4_, 0.00375 M NaN_3_, and 0.375 M KF; pH 7.4). Homogenization of samples was conducted using Qiagen TissueLyser LT (Qiagen Inc.) with a 5-mm-diameter steel bead for 10 min at 50 Hz. Homogenized samples were centrifuged for 5 min at 14,000× *g*. Supernatants were filtered through a 3000 MWCO Amicon Ultra-0.5 filter (Millipore Sigma, Oakville, ON, Canada) by centrifugation at 14,000× *g* for 30 min at 4 °C. An aliquot of 360 µL was mixed with metabolomics buffer (200 µL) and deuterium oxide (140 µL) containing 0.05% *v*/*v* trimethylsilylpropanoic acid (TSP) for a final volume of 700 µL. TSP was used as a chemical shift reference for ^1^H-NMR spectroscopy. Resulting solutions were vortexed, centrifuged at 12,000× *g* for 5 min at 4 °C, and an aliquot of 550 µL of each supernatant was loaded in a 5 mm NMR tube. Spectra obtained from each sample were processed as previously described [24].

MATLAB (MathWorks, Natick, MA, USA) was used to bin and align spectral peaks through Dynamic Adaptive Binning [74] and Recursive Segment Wise Peak Alignment [75], respectively. Data was normalized to the total metabolome and pareto-scaled to reduce influence of intense peaks. Metabolites were identified and quantified using Chenomx NMR Suite 8.5 standard (Edmonton, AB, Canada). The concentration of each metabolite was calculated using the known internal concentration of the TSP peak for each sample (0.37 mM). To determine significant differences between treatments, spectral bins were subjected to univariate and multivariate analysis. Univariate measures were obtained with MATLAB using the decision tree algorithm outlined by Goodpaster et al. [76]. Multivariate tests used a Variable Importance Analysis based on random Variable Combination (VIAVC) algorithm. The VIAVC algorithm combines Partial Least Squares Discriminant Analysis (PLS-DA), and area under the Receiving Operating Characteristics (ROC) curve, to determine metabolites for group classification [77]. The *p*-values obtained were Bonferroni-Holm corrected to account for multiple comparisons. Percent differences of bins between treatments were calculated in MATLAB. MetaboAnalyst R package (v. 3.0) was used for orthogonal partial least squares discriminant analysis (OPLS-DA), and to determine pathways associated with relevant metabolites [78,79]. In all instances, Q^2^ values presented in the results section were significant at *p* ≤ 0.058.

## 5. Conclusions

We evaluated the progression of salmonellosis in broiler chicks by characterizing histopathologic changes, host immune response, impacts on the host metabolome, and alterations in function of the enteric microbiota. We observed that early infection of neonatal broiler chicks by *S*. Typhimurium had a negative impact on the birds by modifying metabolic processes, and altering the function, but not the structure of the enteric microbiota. Collectively evaluating the host immune response and the metabolome showed that the pro-inflammatory response incited by the pathogen affected every organ examined, including breast muscle and hippocampus. Moreover, results illustrated the importance of preventing early colonization of chicks by *S.* Typhimurium to ameliorate the deleterious impacts of an immune response on the host and function of the enteric microbiota, both of which can negatively affect production health (e.g., increased catabolic cost to mounting an inflammatory response). The identification and use of metabolite biomarkers of disease as a diagnostic tool (i.e., detectible before disease is manifested) would be of value to poultry producers, and this is the subject of our current research.

## Figures and Tables

**Figure 3 pathogens-11-01257-f003:**
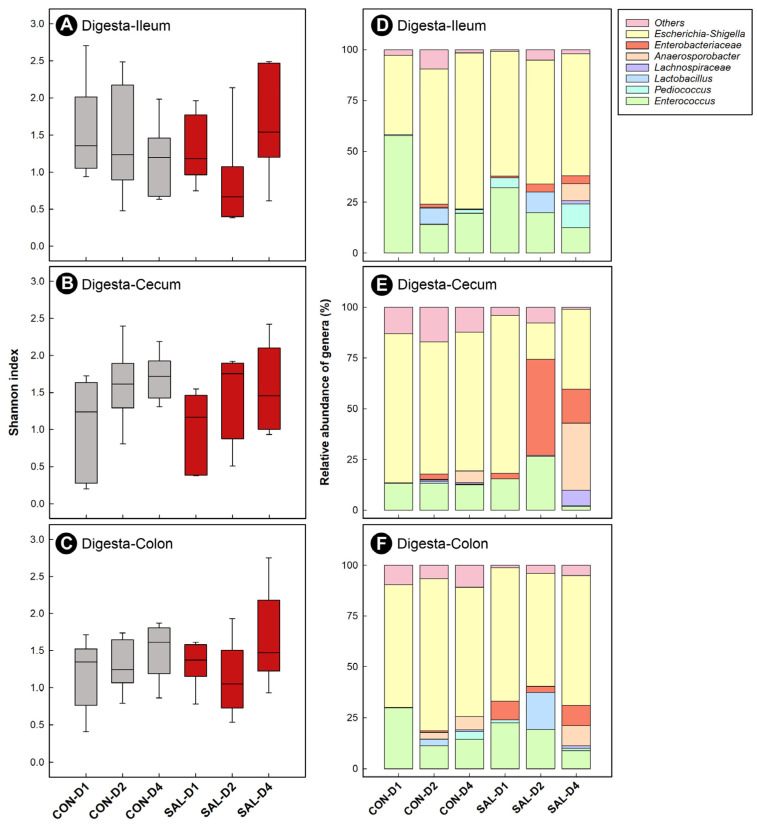
Alpha diversity of bacterial communities and relative abundance of taxa in digesta of broiler chicks inoculated with *Salmonella enterica* serovar Typhimurium (SAL) or administered medium alone (CON). Samples were obtained at 1 (D1), 2 (D2), and 4 (D4) days post-inoculation. (**A**–**C**) Alpha diversity in digesta. (**A**) Ileum. (**B**) Cecum. (**C**) Colon. Whisker plots represent interquartile ranges, and horizontal black lines indicate the median value; the size of the boxes denote the distribution within a confidence of 95%, and vertical lines represent the total distribution of the data. (**D**–**F**) Relative abundance of genera in digesta. (**D**) Ileum. (**E**) Cecum. (**F**) Colon.

**Figure 4 pathogens-11-01257-f004:**
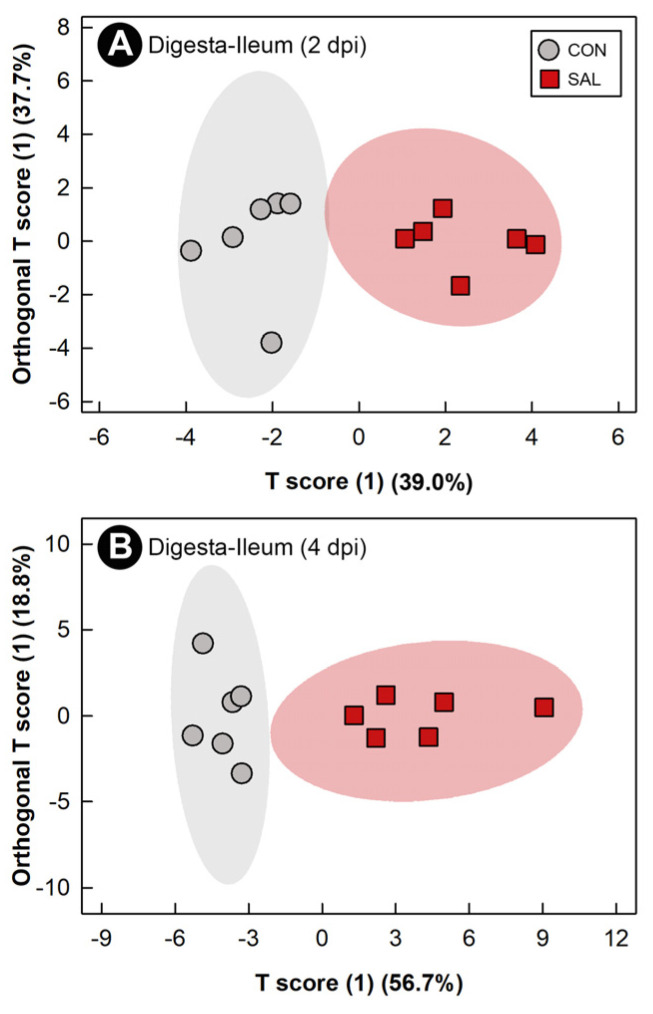
Orthogonal Projections to Latent Structures Discriminant Analysis (OPLS-DA) scores plots of ileal digesta metabolites in broiler chicks inoculated with *Salmonella enterica* serovar Typhimurium (SAL) or administered medium alone (CON). (**A**) 2 days post-inoculation (dpi). (**B**) 4 dpi. Each circle or square represents one bird, and data were plotted using significant metabolites identified by Mann–Whitney U test and Variable Importance Analysis based on Variable Combination (VIAVC) machine learning. Shaded ellipses represent 95% confidence intervals.

**Figure 5 pathogens-11-01257-f005:**
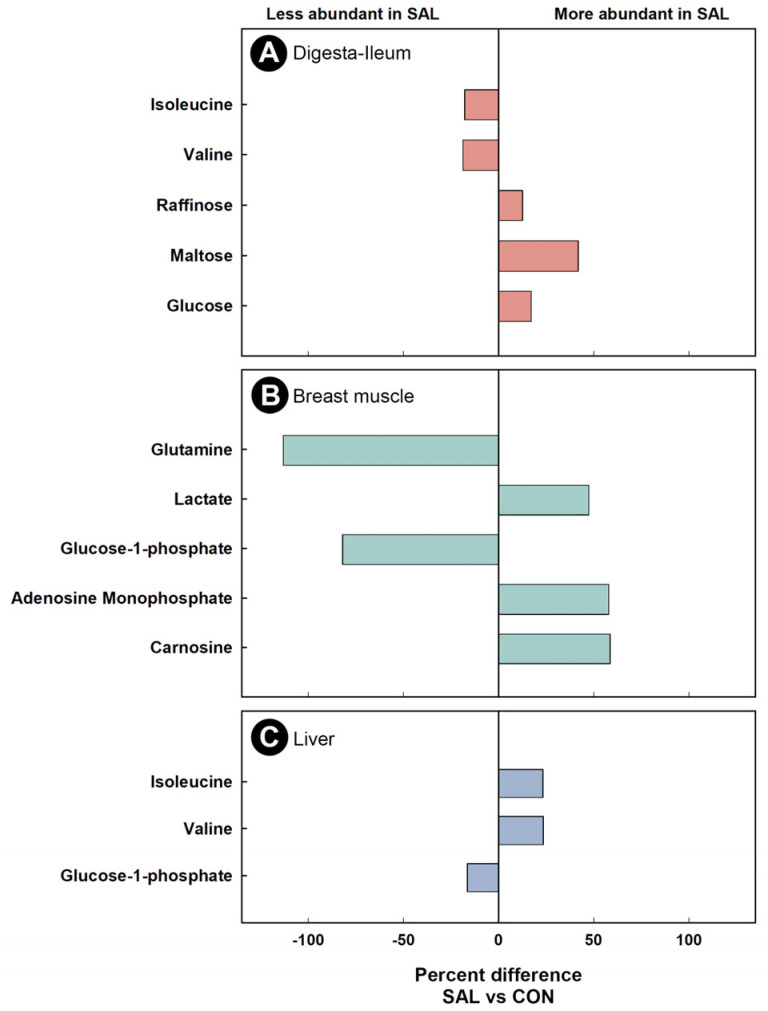
Percent difference of discriminated metabolites in broiler chicks inoculated with *Salmonella enterica* serovar Typhimurium (SAL) or administered medium alone (CON) at 4 days post-inoculation. Only metabolites that were differentially abundant between SAL and CON treatment birds are shown. (**A**) Ileal digesta. (**B**) Breast muscle. (**C**) Liver.

## Data Availability

The microbiota raw sequencing reads were submitted to the Sequencing Read Archive of NCBI under BioProject accession number PRJNA876289.

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
