# Peer review of "Infection by Salmonella enterica Serovar Typhimurium DT104 Modulates Immune Responses, the Metabolome, and the Function of the Enteric Microbiota in Neonatal Broiler Chickens"

_pathogens, 2022, doi:10.3390/pathogens11111257_

Round 1

Reviewer 1 Report

In this study, Bescucci et al examined the immune responses and the microbial community in the ileum, cecum, and colon, and metabolome of digesta, serum, liver, hippocampus, and breast muscle after infecting neonatal broiler chickens orally with S. Typhimurium DT104. This study is informative and important in the field. However, following comments will be helpful to further improve the quality of the manuscript.

11. This study was focused to examine the early changes of the disease due to Salmonella infection and included three time points of post infections ( 1 dpi, 2 dpi, 4 dpi). By looking at the data I felt this study will be more informative if investigators include later time points such as 7 dpi or 14 dpi. In this infection model, do birds survive Salmonella infection naturally? If so, how long does it take to clear the symptoms?  For example, densities of Salmonella from digesta and associated with mucosa from the ileum, cecum, and colon of infected broiler chicks at 1, 2, and 4 dpi were similar. These findings indicate that during the infection period examined birds were not able to clear the Salmonella infection. Similarly, authors noted a little or no changes in microbial communities in these organs/tissues. Would it be possible to changes in microbiota at later time course and the microbiota shift might be beneficial to protect hosts from Salmonella infection?

22.  For simplicity, please indicate the panels of each figure with the post infection time and organ (e.g. cecum at 1 dpi).

33.   In section 2.2 Line 111-114: Please include the body weight data in the supplementary materials.

44. In section 2.4 Line 126-126: They mentioned that expression of hallmark immune genes were examined at 1 dpi, 2 dpi, and 4 dpi in the ileum, cecum, and colon, however; they did not mention or include the data at 1 dpi. Please mention in the result and include the data at 1 dpi in the supplementary materials.

55. Whenever appropriate please indicate statistical analysis methods used (e.g. student’s t-test, ANOVA etc.).

Author Response

Reviewer #1

General Comments

  1. Reviewer comment. In this study, Bescucci et al examined the immune responses and the microbial community in the ileum, cecum, and colon, and metabolome of digesta, serum, liver, hippocampus, and breast muscle after infecting neonatal broiler chickens orally with S. Typhimurium DT104. This study is informative and important in the field. However, following comments will be helpful to further improve the quality of the manuscript.

Author response. No response required.

Specific comments

  1. Reviewer comment. This study was focused to examine the early changes of the disease due to Salmonella infection and included three time points of post infections ( 1 dpi, 2 dpi, 4 dpi). By looking at the data I felt this study will be more informative if investigators include later time points such as 7 dpi or 14 dpi. In this infection model, do birds survive Salmonella infection naturally? If so, how long does it take to clear the symptoms? For example, densities of Salmonella from digesta and associated with mucosa from the ileum, cecum, and colon of infected broiler chicks at 1, 2, and 4 dpi were similar. These findings indicate that during the infection period examined birds were not able to clear the Salmonella infection. Similarly, authors noted a little or no changes in microbial communities in these organs/tissues. Would it be possible to changes in microbiota at later time course and the microbiota shift might be beneficial to protect hosts from Salmonella infection?

Author response. Determination of sampling days was based on the literature (see response to Reviewer #2 Specific Comment #2). Berndt et al. [1] previously demonstrated that copy numbers of S. Typhimurium, and the immune response directed to control the infection increases from day 1 to day 4, being the latter, the point at which infection peaks. The Berndt et al. study also demonstrated that infection starts resolving after day 4. Moreover, they observed no changes in histopathologic scores, immune responses, or copy numbers of the pathogen 4 hours after inoculation. In a production setting this will probably be expedited due to the acquisition of a more diverse microbiota (i.e. from the environment). In this regard, a more diverse microbiota will better compete with the Salmonella enterica serovar Typhimurium for nutrients resulting in colonization resistance. Consistent with this possibility, other studies conducted in older chickens experimentally infected with S. Typhimurium did not show pathology [2], and a recently conducted study showed that transplantation of an enteric microbiota conferred protection against the pathogen [3]. Based on the reports in the literature we decided that day 1, day 2, and day 4 were the best sampling times to characterize changes in the metabolome, in immune responses, and in the autochthonous microbiota incited by the pathogen in neonatal broiler chicks.

  1. Reviewer comment. For simplicity, please indicate the panels of each figure with the post infection time and organ (e.g. cecum at 1 dpi).

Author response. Labels have been added to figures as suggested.

  1. Reviewer comment. In section 2.2 Line 111-114: Please include the body weight data in the supplementary materials.

Author response. A figure has been added to the supplementary materials (i.e. Figure S2).

  1. Reviewer comment. In section 2.4 Line 126-126: They mentioned that expression of hallmark immune genes were examined at 1 dpi, 2 dpi, and 4 dpi in the ileum, cecum, and colon, however; they did not mention or include the data at 1 dpi. Please mention in the result and include the data at 1 dpi in the supplementary materials.

Author response. No significant differences were observed among treatments in any of the immune markers of disease evaluated at this sample point. A statement has now been added to the results section. Since no changes were observed, we feel that the addition of a graph in the supplementary section is not necessary.

  1. Reviewer comment. Whenever appropriate please indicate statistical analysis methods used (e.g. student’s t-test, ANOVA etc.).

Author response. Statistical tests were conducted in R, and a description of the analyses conducted are specified in the text. We have highlighted relevant text in the revised marked version of the manuscript.

REFERENCES

  1. Berndt, A.; Wilhelm, A.; Jugert, C.; Pieper, J.; Sachse, K.; Methner, U. Chicken cecum immune response to Salmonella enterica serovars of different levels of invasiveness. Infect Immun 2007, 75, 5993-6007.
  2. Barrow, P.A.; Huggins, M.B.; Lovell, M.A.; Simpson, J.M. Observations on the pathogenesis of experimental Salmonella Typhimurium infection in chickens. Res Vet Sci 1987, 42, 194-199.
  3. Pottenger, S.; Watts, A.; Wedley, A.; Jopson, S.; Darby, A.C.; Wigley, P. Timing and delivery route effects of cecal microbiome transplants on Salmonella Typhimurium infections in chickens. 2022.
  4. Brown, C.L.J.; Zaytsoff, S.J.M.; Montina, T.; Inglis, G.D. Corticosterone-mediated physiological stress alters liver, kidney, and breast muscle metabolomic profiles in chickens. Animals (Basel) 2021, 11.
  5. Bescucci, D.M.; Clarke, S.T.; Brown, C.L.J.; Boras, V.F.; Montina, T.; Uwiera, R.R.E.; Inglis, G.D. The absence of murine cathelicidin-related antimicrobial peptide impacts host responses enhancing Salmonella enterica serovar Typhimurium infection. Gut Pathog 2020, 12, 53.
  6. Metzler-Zebeli, B.U.; Siegerstetter, S.C.; Magowan, E.; Lawlor, P.G.; O'Connell, N.E.; Zebeli, Q. Feed restriction reveals distinct serum metabolome profiles in chickens divergent in feed efficiency traits. Metabolites 2019, 9.
  7. Adhikari, P.; Yadav, S.; Cosby, D.E.; Cox, N.A.; Jendza, J.A.; Kim, W.K. Research note: Effect of organic acid mixture on growth performance and Salmonella Typhimurium colonization in broiler chickens. Poult Sci 2020, 99, 2645-2649.
  8. Smulders, T.V. The avian hippocampal formation and the stress response. Brain, behavior and evolution 2017, 90, 81-91.
  9. Berliner, L.J.; Khramtsov, V.; Fujii, H.; Clanton, T.L. Unique in vivo applications of spin traps. Free Radic Biol Med 2001, 30, 489-499.

Reviewer 2 Report

In the manuscript entitled “Infections by Salmonella enterica serovar Typhimurium DT104 modulates immune responses, the metabolome, and function of the enteric microbiota in neonatal broiler chickens” the authors investigated acute salmonellosis in broiler chicks by infection via oral route. Below are the suggestions to improve the manuscript.

1.    Salmonella enterica serovar Typhimurium should be abbreviated as Salmonella typhimurium or S. typhimurium throughout the manuscript.

2.    What is the rationale for the inclusion of day 1, 2 and 4 days-post inoculation (DPI)? This should be explained in detail. The authors should have included day 0, and an additional day 7 time point in their study.

3.    Why the authors chose breast muscle, liver, blood, and hippocampus for their metabolomics study? The authors should have included gall bladder for metabolomics.  

4.    How many birds were included per group? The authors should include a flow diagram depicting the experimental design and groups of birds used.

5.    The authors should include representative histopathology pictures.

Author Response

Reviewer #2

General Comments

  1. Reviewer comment. In the manuscript entitled “Infections by Salmonella enterica serovar Typhimurium DT104 modulates immune responses, the metabolome, and function of the enteric microbiota in neonatal broiler chickens” the authors investigated acute salmonellosis in broiler chicks by infection via oral route. Below are the suggestions to improve the manuscript.

Author response. No response required.

Specific comments

  1. Reviewer comment. Salmonella enterica serovar Typhimurium should be abbreviated as Salmonella typhimurium or S. typhimurium throughout the manuscript.

Author response. Where appropriate we have replaced Salmonella enterica serovar Typhimurium with Salmonella Typhimurium or S. Typhimurium as suggested.

  1. Reviewer comment. What is the rationale for the inclusion of day 1, 2 and 4 days-post inoculation (DPI)? This should be explained in detail. The authors should have included day 0, and an additional day 7 time point in their study.

Author response. Determination of sampling days was based on the literature (also see response to Reviewer #1 Specific Comment #1). Berndt et al. [1] previously demonstrated that copy numbers of S. Typhimurium, and the immune response directed to control the infection increases from day 1 to day 4, with the latter being the point at which infection peaks. The Berndt et al. study also demonstrated that infection starts resolving after day 4. Moreover, they observed no changes in histopathologic scores, immune responses, or copy numbers of the pathogen 4 hours after inoculation with the pathogen. Based on the findings of Berndt et al., we decided that day 1, day 2, and day 4 were the best sampling times to characterize changes in the metabolome, in immune responses, and in the autochthonous microbiota incited by the pathogen in neonatal broiler chicks. Based on previously reported studies, we did not consider that the inclusion of a day 0 or day 7 endpoint as it was unlikely to add additional relevant information to our study.

  1. Reviewer comment. Why the authors chose breast muscle, liver, blood, and hippocampus for their metabolomics study? The authors should have included gall bladder for metabolomics.

Author response. The selection of the different tissues to evaluate the metabolome were based on previous publications and literature. Following is our rationale for the tissues that we targeted (i.e. by tissue):

Muscle. A previous study published by our team [4] demonstrated the importance of stress in the metabolome of breast muscle. Additionally, alterations in the metabolome of breast muscle is directly associated with bird productivity. Thus, we considered that evaluating the effect that the pathogen has in this site was important for the poultry industry.

Liver. Salmonellosis can progress into a systemic condition, with the liver being the main organ to which the pathogen will disseminate [2]. Additionally, a previous study conducted by our team in mice showed that the liver metabolome was substantively affected by S. Typhimurium [5]. Thus, we felt that evaluating the metabolome of liver was important to understand the systemic effects of the disease in broiler chicks.

Blood. Characterization of the metabolome of blood was conducted to determine early host responses. Additionally, a recent study conducted by Metzler-Zebeli et al. [6] demonstrated the importance of evaluating serum as indicator of nutritional status of chicken. In this regard, S. Typhimurium has been shown to decrease feed intake in chicks resulting in lower body weight [7]. Thus, blood was evaluated to determine the early effects of the pathogen on the metabolome and on the nutritional status of the bird.

Hippocampus. The hippocampus in birds is involved in spatial memory, and it is particularly sensitive to the adverse effect of stress [8]. A previous study conducted by our team, currently submitted to the journal, Metabolites (Brown et al. in review) showed that several stressors (i.e. heat, isolation, corticosterone) profoundly altered the metabolome of the hippocampus in broiler chicks. Thus, we targeted the hippocampus in an attempt to more fully understand the effect of the pathogen at triggering an acute stress response.

Gall bladder. Since the liver is the primary colonization site we felt that the analysis of the metabolome of the liver was sufficient to cover this aspect of the pathogenesis. However, the reviewer’s comment on including the gall bladder is well taken (i.e. as it is a site of carriage for Salmonella enterica serovar Enteritidis), and in future studies using the S. Typhimurium model of salmonellosis we will also target the gall bladder. We thank the reviewer for this suggestion.

  1. Reviewer comment. How many birds were included per group? The authors should include a flow diagram depicting the experimental design and groups of birds used.

Author response. A total of 36 animals was used for the experiment, and this has been added to the manuscript. Moreover, we have provided clarification on the factorial experimental design used within the text, and added an experimental design figure, which has been added as a supplemental figure (i.e. Figure S5).

  1. Reviewer comment. The authors should include representative histopathology pictures.

Author response. Histopathology pictures are provided in the supplemental material (i.e. Figure S2).

REFERENCES

  1. Berndt, A.; Wilhelm, A.; Jugert, C.; Pieper, J.; Sachse, K.; Methner, U. Chicken cecum immune response to Salmonella enterica serovars of different levels of invasiveness. Infect Immun 2007, 75, 5993-6007.
  2. Barrow, P.A.; Huggins, M.B.; Lovell, M.A.; Simpson, J.M. Observations on the pathogenesis of experimental Salmonella Typhimurium infection in chickens. Res Vet Sci 1987, 42, 194-199.
  3. Pottenger, S.; Watts, A.; Wedley, A.; Jopson, S.; Darby, A.C.; Wigley, P. Timing and delivery route effects of cecal microbiome transplants on Salmonella Typhimurium infections in chickens. 2022.
  4. Brown, C.L.J.; Zaytsoff, S.J.M.; Montina, T.; Inglis, G.D. Corticosterone-mediated physiological stress alters liver, kidney, and breast muscle metabolomic profiles in chickens. Animals (Basel) 2021, 11.
  5. Bescucci, D.M.; Clarke, S.T.; Brown, C.L.J.; Boras, V.F.; Montina, T.; Uwiera, R.R.E.; Inglis, G.D. The absence of murine cathelicidin-related antimicrobial peptide impacts host responses enhancing Salmonella enterica serovar Typhimurium infection. Gut Pathog 2020, 12, 53.
  6. Metzler-Zebeli, B.U.; Siegerstetter, S.C.; Magowan, E.; Lawlor, P.G.; O'Connell, N.E.; Zebeli, Q. Feed restriction reveals distinct serum metabolome profiles in chickens divergent in feed efficiency traits. Metabolites 2019, 9.
  7. Adhikari, P.; Yadav, S.; Cosby, D.E.; Cox, N.A.; Jendza, J.A.; Kim, W.K. Research note: Effect of organic acid mixture on growth performance and Salmonella Typhimurium colonization in broiler chickens. Poult Sci 2020, 99, 2645-2649.
  8. Smulders, T.V. The avian hippocampal formation and the stress response. Brain, behavior and evolution 2017, 90, 81-91.
  9. Berliner, L.J.; Khramtsov, V.; Fujii, H.; Clanton, T.L. Unique in vivo applications of spin traps. Free Radic Biol Med 2001, 30, 489-499.

Reviewer 3 Report

Comments for the Author (Required):

The study by Bescucci DM et al, titled “Infection by Salmonella enterica serovar Typhimurium DT104 2 Modulates Immune Responses, the Metabolome, and the Function of the Enteric Microbiota in Neonatal Broiler Chickens aims to understand the impact of Salmonella enterica serovar Typhimurium infection on immune responses, cellular metabolism, and intestinal bacterial diversity in Neonatal Broiler Chickens. Although it is an interesting article, I have some concerns about this article.

Author showed that Salmonella enterica serovar Typhimurium upregulates some of immune genes (Fig 2). It would be nice to quantify circulatory inflammatory molecule by ELISA. 

Since, authors have discussed more about oxidative stress, it would be interesting to quantify reactive oxygen species and reactive nitrogen species. 

Page 5, para run like this- “DNA of this genus was also observed at higher abundance in the colonic digesta of challenged birds at 4 dpi (Figure 3F)”. – it seems like the abundance of Enterobacteriaceae family is high in both 1 dpi as well as 4 dpi. I am curious know why the abundance of Enterobacteriaceae family is low in 2 dpi. On other hand, the abundance of lactobacillus is higher in 2 dpi.  

Page 10, para run like this- “Arrival of inflammatory cells, specifically macrophages and heterophils, results in an increase of reactive oxygen species (e.g., INOS) [26]”. I believe that iNOS increases reactive nitrogen species not reactive oxygen species. – Please reformulate 

Author Response

Reviewer #3

General Comments

  1. Reviewer comment. The study by Bescucci DM et al, titled “Infection by Salmonella enterica serovar Typhimurium DT104 2 Modulates Immune Responses, the Metabolome, and the Function of the Enteric Microbiota in Neonatal Broiler Chickens” aims to understand the impact of Salmonella enterica serovar Typhimurium infection on immune responses, cellular metabolism, and intestinal bacterial diversity in Neonatal Broiler Chickens. Although it is an interesting article, I have some concerns about this article.

Author response. The reviewer has identified some points that require clarification/justification, which have all been addressed within the manuscript and/or in our responses to specific comments below. We thank the reviewer for the time expended on reviewing our manuscript.

  1. Reviewer comment. Author showed that Salmonella enterica serovar Typhimurium upregulates some of immune genes (Fig 2). It would be nice to quantify circulatory inflammatory molecule by ELISA.

Author response. We agree with the reviewer that protein quantification provides biological relevance compared to gene expression since proteins are not always produced after transcription. Despite this, gene expression can show that signals preceding mRNA expression were present differentially between control and infected animals. This is indication that a stimulation of the immune pathways was triggered by the pathogen. Although this would not indicate biological active proteins, the aim of our study was to demonstrate that S. Typhimurium stimulated a response by host response.

  1. Reviewer comment. Since, authors have discussed more about oxidative stress, it would be interesting to quantify reactive oxygen species and reactive nitrogen species.

Author response. We agree with the reviewer regarding the quantification of reactive oxygen and nitrogen species. However, to the best of our knowledge, the measurement of these components entails the used of external ‘traps or probes’ to be able to intercept the reactive radicals. The application of a probe in our experiment could have had confounding effects related to the stress response triggered by the pathogen. Additionally, it is well know that radical species have a very short lifespan with many detection systems unable to accurately quantify these molecules in time [9]. As a result, we did not attempt to quantify reactive oxygen species and reactive nitrogen species in the current study. However, the reviewer’s point is well taken, and we will explore this in future research conducted by our team.

  1. Reviewer comment. Page 5, para run like this- “DNA of this genus was also observed at higher abundance in the colonic digesta of challenged birds at 4 dpi (Figure 3F)”. – it seems like the abundance of Enterobacteriaceae family is high in both 1 dpi as well as 4 dpi. I am curious know why the abundance of Enterobacteriaceae family is low in 2 dpi. On other hand, the abundance of lactobacillus is higher in 2 dpi.

Author response. The reviewer raises a very good point. We believe that the lower abundance of bacteria within the Enterobacteriaceae observed at 2 days post-inoculation (dpi) in the ileum and colon was associated with the high abundance of bacteria in this family within the cecum. It is possible that these bacteria were concentrated in the cecum at the time of sampling, since as explained in the discussion, the cecal content is only voided twice per day into the colon. That could also explain the similarity between the ileum and colonic abundance. Even though the Lactobacillus family seems to be in higher abundance at 2 dpi, no significance differences were observed with ANCOM. This is why we did not discuss this result in our manuscript.

  1. Reviewer comment. Page 10, para run like this- “Arrival of inflammatory cells, specifically macrophages and heterophils, results in an increase of reactive oxygen species (e.g., INOS) [26]”. I believe that iNOS increases reactive nitrogen species not reactive oxygen species. – Please reformulate

Author response. The text has been modified.

REFERENCES

  1. Berndt, A.; Wilhelm, A.; Jugert, C.; Pieper, J.; Sachse, K.; Methner, U. Chicken cecum immune response to Salmonella enterica serovars of different levels of invasiveness. Infect Immun 2007, 75, 5993-6007.
  2. Barrow, P.A.; Huggins, M.B.; Lovell, M.A.; Simpson, J.M. Observations on the pathogenesis of experimental Salmonella Typhimurium infection in chickens. Res Vet Sci 1987, 42, 194-199.
  3. Pottenger, S.; Watts, A.; Wedley, A.; Jopson, S.; Darby, A.C.; Wigley, P. Timing and delivery route effects of cecal microbiome transplants on Salmonella Typhimurium infections in chickens. 2022.
  4. Brown, C.L.J.; Zaytsoff, S.J.M.; Montina, T.; Inglis, G.D. Corticosterone-mediated physiological stress alters liver, kidney, and breast muscle metabolomic profiles in chickens. Animals (Basel) 2021, 11.
  5. Bescucci, D.M.; Clarke, S.T.; Brown, C.L.J.; Boras, V.F.; Montina, T.; Uwiera, R.R.E.; Inglis, G.D. The absence of murine cathelicidin-related antimicrobial peptide impacts host responses enhancing Salmonella enterica serovar Typhimurium infection. Gut Pathog 2020, 12, 53.
  6. Metzler-Zebeli, B.U.; Siegerstetter, S.C.; Magowan, E.; Lawlor, P.G.; O'Connell, N.E.; Zebeli, Q. Feed restriction reveals distinct serum metabolome profiles in chickens divergent in feed efficiency traits. Metabolites 2019, 9.
  7. Adhikari, P.; Yadav, S.; Cosby, D.E.; Cox, N.A.; Jendza, J.A.; Kim, W.K. Research note: Effect of organic acid mixture on growth performance and Salmonella Typhimurium colonization in broiler chickens. Poult Sci 2020, 99, 2645-2649.
  8. Smulders, T.V. The avian hippocampal formation and the stress response. Brain, behavior and evolution 2017, 90, 81-91.
  9. Berliner, L.J.; Khramtsov, V.; Fujii, H.; Clanton, T.L. Unique in vivo applications of spin traps. Free Radic Biol Med 2001, 30, 489-499.

Round 2

Reviewer 1 Report

All the comments have been addressed satisfactorily.